behaviour, neuroscience, evolution

acoustic communication, decision making, sex-specific, grasshopper

**Author for correspondence:**
Jan Clemens
e-mail: clemensjan@gmail.com

# Sex-specific speed–accuracy trade-offs shape neural processing of acoustic signals in a grasshopper

Jan Clemens[1], Bernhard Ronacher[2] and Michael S. Reichert[3]

[1]European Neuroscience Institute Göttingen – A Joint Initiative of the University Medical Center Göttingen and the Max-Planck Society, Grisebachstrasse 5, Göttingen 37077, Germany
[2]Behavioral Physiology Group, Department of Biology, Humboldt-Universität zu, Berlin, Germany
[3]Department of Integrative Biology, Oklahoma State University, Stillwater, OK USA

  JC, 0000-0003-4200-8097; BR, 0000-0001-8640-9007; MSR, 0000-0002-0159-4387

Speed–accuracy trade-offs—being fast at the risk of being wrong—are fundamental to many decisions and natural selection is expected to resolve these trade-offs according to the costs and benefits of behaviour. We here test the prediction that females and males should integrate information from courtship signals differently because they experience different pay-offs along the speed–accuracy continuum. We fitted a neural model of decision making (a drift–diffusion model of integration to threshold) to behavioural data from the grasshopper *Chorthippus biguttulus* to determine the parameters of temporal integration of acoustic directional information used by male grasshoppers to locate receptive females. The model revealed that males had a low threshold for initiating a turning response, yet a large integration time constant enabled them to continue to gather information when cues were weak. This contrasts with parameters estimated for females of the same species when evaluating potential mates, in which response thresholds were much higher and behaviour was strongly influenced by unattractive stimuli. Our results reveal differences in neural integration consistent with the sex-specific costs of mate search: males often face competition and need to be fast, while females often pay high error costs and need to be deliberate.

## 1. Introduction

Sensory information is inherently noisy. Repeated sampling and integration of information over time reduces noise, and is a ubiquitous strategy in information processing and decision making [1–3]. The neural algorithm by which ecologically relevant sensory information is integrated is expected to be shaped by a fundamental trade-off between the speed of decision making and the accuracy of the inferred sensory input [1,4,5]. Studies of animals trained in artificial decision-making tasks show that the resolution of the speed–accuracy trade-off depends on the relative costs of delaying the decision to accumulate more information compared to the costs of making an error based on insufficient information, the signal-to-noise ratio and the stability of the sensory information over time [6–12]. However, all of these factors vary in natural environments, and among individuals, sexes and species [3,13,14]. While variation in speed–accuracy trade-offs with condition or experience has been reported for naturalistic tasks [15–19], there exists little direct evidence that natural integration processes are shaped by selection, for instance from comparisons across groups expected to face different costs [3,20].

The processing of signals related to mate choice presents a clear instance in which selection likely favours different resolutions of the speed–accuracy trade-off in the two sexes, which in turn are expected to result in sex differences in

temporal integration: integration processes in males should facilitate the fast decisions required for successful competition, while integration in females should be slower, but enable more accurate decisions about male quality. The existence of sex-specific circuits in the nervous system suggests that temporal integration could indeed be implemented in a sex-specific manner [21,22], but whether the characteristics of temporal integration differ between males and females in a natural task is unclear. Here we combine an existing behavioural dataset [23] with new data and fit a drift–diffusion model (DDM) [5] to characterize how the nervous system accumulates sensory cues and triggers decisions in mate searching. Based on the expected costs and benefits of different integration strategies under sexual selection theory, we test predictions for how integration may differ between males and females evaluating acoustic signals of the opposite sex.

The grasshopper *Chorthippus biguttulus* provides an excellent model for studies of temporal integration because both males and females produce and respond to acoustic signals during mate searching (figure 1a) [24], but integration and decision making strategies are expected to differ between the sexes because they are subject to different selection pressures [25]. Males produce calling songs to find females; receptive females are stationary but respond with songs that facilitate mate localization by the male [26]. In *C. biguttulus*, females pay high costs from making errors [25,27,28], because they are egg-limited and mating with a male of another species or of low genetic quality produces no or low-quality offspring. In addition, singing exposes females to predators and parasitoids. Females should therefore avoid responding to the song from males of another species or of low genetic quality. By contrast, females do not face competition from other females and therefore are not under pressure to be fast. Females are therefore expected to favour accuracy over speed when evaluating the male song pattern. This was confirmed by a DDM for temporal integration based on female response behaviour [29,30]. The model parameters indicated that females integrate information across the entire calling song of a male with a high threshold for response and very high negative weighting of unattractive song components (i.e. those of heterospecific or malformed males). These integration parameters ensure the accurate detection of unsuitable males combined with slow behavioural responses to attractive males.

Here, we extend the modelling approach to male mate localization, which is predicted to have very different integration characteristics. Female density is low in the visually cluttered environment and females lack conspicuous visual characteristics or long-range chemical cues that would allow males to find them. Chance encounters are therefore rare and the female response song is often the only possibility for localizing receptive females [28,31]. Furthermore, the speed of approach is critical because females already engaged in close-range courtship with faster arriving males will not continue to advertise their position, preventing slower males from localizing those females. Thus, although males rarely directly interact with one another in physical competitions, they nevertheless face high levels of competition to rapidly localize responsive females in a crowded and noisy environment [32]. Males should therefore favour speed over accuracy to a greater degree than females, although we do not expect males to completely disfavour accuracy when

integrating directional cues from the female song, since localization errors increase the time exposed to predators and parasitoids, and will prevent them from finding the female. We predict therefore that males will have higher sensory weightings (or equivalently, a lower response threshold) than females, reflecting their speedier response. This puts them at risk of making errors if early sensory information is wrong. Furthermore, to maintain accuracy when directional cues are equivocal, we predict that integration times will be at least as long as typical female songs so that males can maximize the chances of integrating sufficient directional information from female signals.

To test these predictions, we used new and previously published behavioural data from a two-speaker playback design that measured male localization of artificial female songs with conflicting directional cues [23]. We applied a DDM to the behavioural data to determine the parameters of temporal integration in males. The DDM corresponded very well with males' decisions, and the model's parameters matched our predictions of long temporal integration times and a low threshold for response, which contrasts with the parameters determined for female behaviour using the same model. This reveals sex-specific differences in the neural processing of sexual signals consistent with predictions from sexual selection theory.

## 2. Methods

### (a) Animals

Behavioural data were collected as described in [23]. We used laboratory-reared and wild-caught males of the species *C. biguttulus*. Laboratory-reared males were the F1 offspring from wild-caught individuals and were isolated by sex at the last instar nymph stage and reared in cages separated by sex. Wild-caught males could have mated previously but were kept separate from females for at least 3 days before the experiments, which is sufficient for them to regain motivation to respond to female signals [33,34]. To further minimize variation in male motivation to respond, we only tested males that responded to a test signal from an attractive female, indicating high motivation to engage in courtship behaviour. We did not systematically track male age or exclude males based on their age. Both laboratory-reared and wild-caught males were group-housed in male-only cages in similar conditions. In group housing, all males would have been exposed to the song of other males, as well as the songs of female playbacks used to assess male motivation and identify test subjects. We, as well as previous studies [35], did not observe any systematic differences in the selectivity of laboratory-reared or wild-caught males, nor any effect of previous experience on the integration of acoustic cues from females. There is also no evidence for learning in this communication system.

### (b) Behavioural experiments

Motivated adult males were placed between two speakers that broadcast an artificial female song stimulus (figure 1b). A female song consists of subunits (syllables) that are separated by pauses. The syllables in our female model song were separated by a 17.5 ms pause; each syllable consisted of six sound pulses (average pulse duration 10.7 ms). This stimulus pattern was highly attractive and reliably elicited turning responses in males, allowing us to assess how directional cues from the stimulus were integrated by the males. Syllables that lack a pause or do not consist of distinct sound pulses are not attractive to males

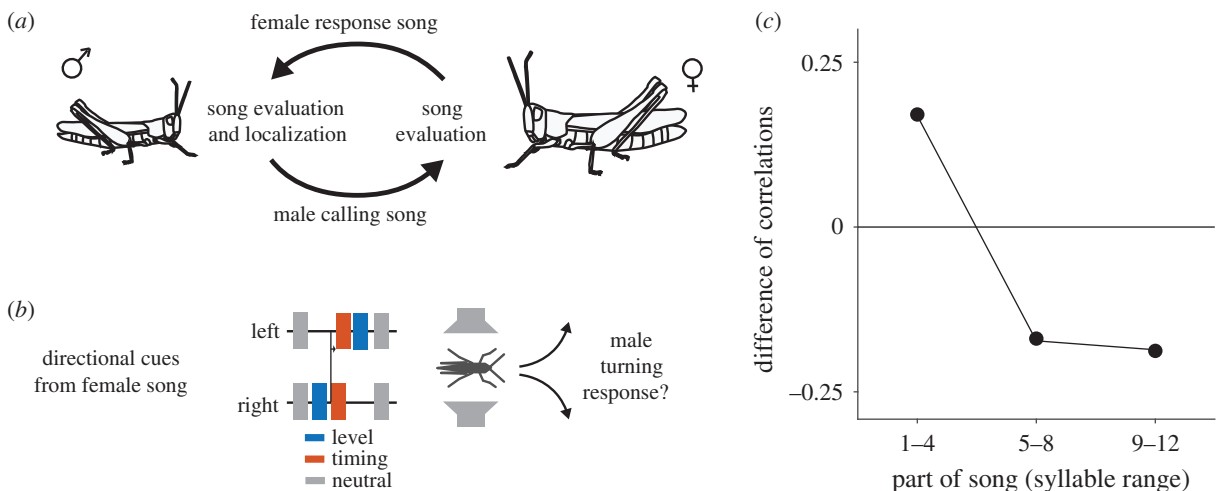

**Figure 1.** Performance of a simple averaging model compared to male behaviour. (*a*) Bidirectional acoustic communication during mate search in the grasshopper *C. biguttulus*. (*b*) Schematic of the paradigm—two speakers were placed on either side of the male, artificial female song is played and the direction of the male turning response is scored. Directional cues are provided by each syllable and arise from level differences (sound on one speaker only, blue) or timing differences (sound on one speaker delayed by 4 ms, red). Both cue types are known to elicit turning responses in males. (*c*) Difference in the correlation of different thirds of the 12-syllable songs observed in behaviour and estimated from the stimulus statistics. The beginning tends to be more, the middle and end less influential on behaviour than expected from the stimulus statistics. See electronic supplementary material, figure S2 for details and number of stimuli. (Online version in colour.)

and fail to elicit male turning responses [36–38]. Individual syllables were manipulated to have timing or level differences between the speakers (see below). Stimuli were broadcast at 60 dB SPL at the position of the male. Males were presented with 10 repetitions of each stimulus. We tested a total of 204 males and most males were tested with more than one stimulus. The median number of different stimuli tested per male (with 10 repetitions per stimulus) was eight (inter-quartile range 5–10). Stimuli were repeated at a variable rate because each time the male moved, we had to re-position the speakers to centre the male once he was again stationary. Forty-four stimuli contained 12 syllables to mimic a typical female song, but we also tested shortened stimuli with eight (three stimuli), five (33 stimuli) or three (one stimulus) syllables to better characterize the dynamics of integration. We include data for male responses from a total of 81 stimuli (electronic supplementary material, figure S1). Data for 38/81 stimuli were previously published in [23].

Male lateralization behaviour was quantified as follows. First, the response for each male was quantified as the proportion of turns directed towards the speaker designated as the reference out of the total number of turns towards either speaker ('0' if the male turned away from the reference speaker, '1' when the male turned towards the reference speaker). For some stimuli, males responded to the stimulus but moved forward instead of towards one of the speakers and we scored these responses as '0.5', equivalent to a decision probability of 0.5 towards (score 1.0) and 0.5 away (score 0.0) from the reference speaker. We then averaged the responses across all males tested with that stimulus (*N* = 15–23 males tested per stimulus, median 20 males). The experimental set-up did not allow us to score turning latencies and those data were therefore not available for model fitting. However, our stimulus design, with conflicting cues placed in different positions within the song, allows us to reliably infer the dynamics of cue integration from the response scores (see below).

## (c) Stimulus design

The dynamics of sensory integration in males were inferred using artificial female songs that varied in duration and in the sequence of directional cues (see electronic supplementary material, figure S1 for all stimulus patterns). Each syllable provided one of three types of directional cue: (i) level cues: we

generated stimuli with level differences by silencing some syllables on one speaker channel. This effectively results in an 8 dB interaural level difference [39,40], (ii) Timing cues, in which the syllable from one speaker led the other by 4 ms or (iii) no directional cues (neutral), in which syllables were presented simultaneously at equal amplitude from both channels. Both timing and level cues elicit orientation responses in male grasshoppers, but are expected to provide directional cues of different strength depending on the magnitude of each cue. We did not attempt to equalize the strength of the timing and level cues used for our stimuli, and instead estimated these parameters from the models. We systematically varied the number and location within the song of these directional cues to generate stimuli with different amounts of directional information, and in some cases with conflicting directional information. This stimulus design with serially conflicting directional cues was critical for calibrating the model parameters [41]. For instance, responses to songs in which syllables at the beginning of the song indicated a female in the direction of one speaker and those at the end of the song indicated a female in the opposite direction reveal over how many syllables males integrate and when decisions are fixed. Combining this stimulus design with a neural model of decision making (see below) allowed us to infer the sensory weights and thresholds, and estimate decision times, even in the absence of reaction time data. A control stimulus with neutral directional cues elicited turning responses with random directions (score 0.53, random turning would produce 0.5). Another control stimulus that was broadcast from only one speaker, reliably elicited turns in males (90% of trials) and all of those turns were correctly directed towards the broadcasting speaker. See electronic supplementary material, figure S1 for a list of all stimulus patterns used in this study.

## (d) Comparison of correlations between the stimulus and the behaviour

We assessed the strength of the relationship (squared Pearson's r) between the males' turning responses and different parts of the 12 syllable stimuli, by dividing each stimulus in thirds (syllables 1–4, 5–8, 9–12), calculating the average directional cue for each third and correlating that average with the males' turning responses (electronic supplementary material, figure S2B). To

account for stimulus-intrinsic correlations, arising from regularity in the stimulus sequences (electronic supplementary material, figure S1), we also correlated the average cue for each third with the average cue over the full song (electronic supplementary material, figure S2A). This revealed that the middle of the song was most strongly correlated with the cues from the full song, which is a result of our stimulus design because the cue direction often changed halfway through the song (electronic supplementary material, figure S1). The difference of the correlation obtained from the behaviour and from the full stimulus indicates stimulus thirds that are more or less influential on the behaviour than expected from the stimulus statistics (electronic supplementary material, figure S2C, figure 1c).

### (e) Modelling
The stimulus is defined as a sequence $s(t)$ with $t = [1, \dots , T]$, $T$ being the number of syllables in the stimulus and a sign indicating the side of the cue relative to the reference speaker ('−1' away from the reference, '+1' towards the reference, '0' neutral).

#### (i) Averaging model
As a baseline, the averaging model simply averages the directional information over the full song and the predicted response $\rho$ is then a function of that average: $\rho = f(x(T+1))$, with $x(t+1) = x(t) + Z(s(t))$, $x(0) = 0$. The sign function $Z(s(t))$ returns −1 if $s(t) < 0$ and +1 otherwise. To account for saturation effects, we set $f$ to be a sigmoidal, which was fitted to minimize the mean-squared error over all stimuli between the prediction $\rho$ of the averaging model and the males' turning response r. However, this only marginally increased the performance of the simple averaging models ($r^2$ linear: 0.72, $r^2$ sigmoidal: 0.75).

#### (ii) Drift–diffusion model
In a DDM, the cues from each syllable are weighted and assigned a sign based on the direction they indicate. The weighted cues are then integrated with an integration timescale $\tau$, which determines the 'leakiness' of integration, with a value of infinity corresponding to perfect integration with no forgotten information, and smaller values corresponding to forgetting of information that came before that time interval. Noise $\sigma$ is added to the integrated sensory information from each syllable, and the decision is fixed when a decision threshold of either $+\theta$ or $-\theta$ is crossed, indicating the decision to turn towards or away from the reference speaker, respectively. If the threshold is not crossed before the end of the song, the decision is made based on the sign of the integrated information at the end of the song. An urgency gain parameter was included to account for the possibility that sensory weights increase or decrease over time [6,42]; increased urgency may be expected for males that need to localize receptive females quickly upon receipt of evidence that one is present. More precisely, the integrated information $x$ after syllable $t$ is given by

$$x(t+1) = \begin{cases} -\theta, & \text{if} & x(t) < -\theta \\ \theta, & \text{if} & x(t) > \theta \\ \dfrac{x(t)}{\tau} + w(t)s(t) + \eta(t)\sigma, & \text{otherwise} \end{cases},$$

with $x(0) = 0$, an integration time constant $\tau$, and a decision threshold $\theta$. Noise $\eta(t)$ was drawn at each time step from a normal distribution with zero mean and unit variance. For timing cues, $w$ was fixed to 1.0 for all models. For level cues, $w$ was fixed to 1.0 for models that did not differentially weight timing and level cues ('single cue' in electronic supplementary material, table S1) and optimized during fitting for models that did (two cues). For models with urgency gain, the sensory weight changed over time and was defined as $w(t) = w*(1 + (t - 1)\gamma)$, with $\gamma$ being the urgency gain [42]. Experiments with

alternative implementations in which the urgency gain reduced the threshold over time [6] yielded similar results of negligible gain. The decision threshold $\theta$ was sticky—once it was crossed, integration ceased and $x(t)$ was fixed to $\pm\theta$. The predicted response, $\rho$, was determined by the sign of the integrated information after the last syllable, $Z(x(T+1))$ averaged over 1000 different instantiations of the noise $\eta$. The simple averaging model can be considered a special case of a DDM with $w = 1$ for level and timing cues, $\sigma = 0$, $\tau = \infty$ and $\theta = \infty$.

#### (iii) Model fitting and evaluation
The parameters of the DDMs were optimized by minimizing the mean-squared error between the predicted and the males' responses using a genetic algorithm ([43], see [44] for details). To speed up convergence, upper and lower bounds were defined for all parameters: $0 < w_{L} < 10$, $1 < \tau < 40$, $0 < \sigma < 5$, $0 < \theta < 10$, $0 < \gamma < 10$. We ensured that these bounds did not affect the final parameter estimates. Fits were evaluated using leave-one-out cross-validation. That is, the model parameters were fitted on all but one stimulus (and its mirror version) and a prediction was then generated for the left-out stimuli. Doing this for all stimuli resulted in 81 parameter estimates and 157 predictions. The squared Pearson's coefficient of correlation, $r^2$, between the predictions and the males' responses was used to quantify model performance. Different models were compared using Akaike's information criterion (AIC), which penalizes models with many parameters. The AIC score is given by $\text{AIC} = 2k + n \ln(e)$, where $k$ is the number of parameters of the model, $n$ is the number of samples used for fitting the model and $e$ is the sum of squared residuals between the predicted and the male's responses: $e = \sum (r - \rho)^2$. Smaller AIC scores are better.

## 3. Results
### (a) Noisy integration to threshold explains turning behaviour
Based on the correlation of the cues in different parts of the song with behaviour, we find that the beginning of the song influences behaviour more than expected (figure 1c, electronic supplementary material, figure S2). This suggests an integration process in males that does not always consider information from the full female song but instead fixes decisions rapidly and dynamically with the available sensory evidence [23]. To account for this finding, we fitted different models of cue integration and decision making. Model comparison (AIC) revealed that the simplest model that explained our data is a DDM with an infinite integration time in which timing and level cues had different weights, but their weights did not change over time (i.e. an urgency gain of zero) (electronic supplementary material, table S1 and figure S3, figure 2a–c). We consider this as the best fit model in discussions below. A threshold-less model that simply averaged directional cues with identical weights for both cue types across the entire song performed worse (figure 2c) as did a DDM variant with identical weights for both cue types (electronic supplementary material, table S1). These simpler models performed well on average (electronic supplementary material, table S1) because for many stimuli in our dataset, the average cue still predicted the behaviour well. However, the performance gap between these models and our best fit model was much higher for stimuli with conflicting or mixed cues, for which correct weighting and stopping of integration after threshold crossing were crucial model

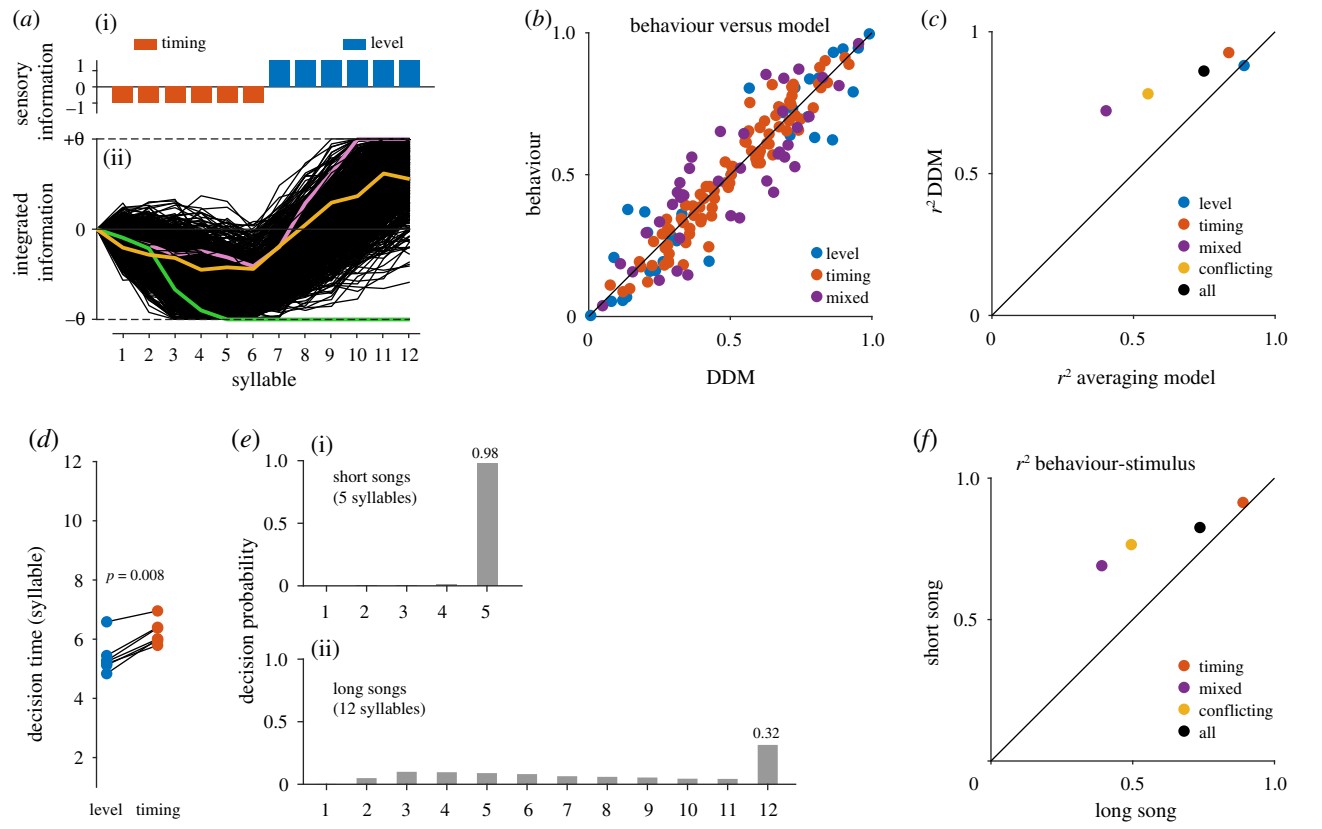

**Figure 2.** A drift–diffusion model (DDM) reproduces the behaviour well and reveals dynamics of temporal integration. (*a*) DDM responses for a 12-syllable stimulus (i) with the first six syllables containing timing cues (red) away from the reference speaker (−), and six syllables with level cues (blue) towards the reference speaker (+). Each cue type is assigned a weight (height of bars, i). Stimulus information is integrated noisily and a decision towards the reference (+) or opposite (−) speaker is fixed when the decision threshold $\theta$ is crossed. Thin black lines indicate 1000 runs with independent noise realizations. Coloured lines highlight example runs that cross the negative threshold (green), the positive threshold (purple) or no threshold (orange), in which case the decision is determined by the sign of the evidence at song end (+). (*b*) Proportion of turns towards the reference speaker in model and behaviour. Colour indicates cue composition of the songs. Diagonal line corresponds to perfect match between model and behaviour. All points are close to that line ($r^2 = 0.86$). (*c*) $R^2$ between model predictions and behavioural data for the best fitting model (DDM) compared to that of a simple averaging model for different data subsets (see legend). The best fitting model outperforms the simple averaging model in particular for stimuli with mixed (purple, stimuli containing timing and level cues) and conflicting cues (yellow, stimuli with cues from both sides). (*d*) Mean decision time (syllable at which threshold is crossed) for seven stimuli with matching patterns (lines) but level (blue) or timing (red) cues. Consistent with their higher weight in the model, level cues drive decisions by about one syllable earlier ($p = 0.008$, left-sided sign test). See electronic supplementary material, figure S4 for the decision time distributions for each of the stimuli depicted here. (*e*) Decision times for short songs with five syllables (i, $N = 66$ stimuli) and long songs with 12 syllables (ii, $N = 83$ stimuli). For most long songs, integration reaches threshold before song end. For nearly all short songs, integration fails to cross threshold. Short song mostly contained timing cues (electronic supplementary material, figure S1). Numbers in the last bar indicate the probability of not reaching the threshold for the two stimulus sets. (*f*) Correlation of behaviour with the average directional cue over the full song for short and long songs. The failure of threshold crossing before song end for short songs (e,i) leads to integration over the full song and a higher correlation with the average directional cue. There is no 'level' stimulus set for this analysis since our dataset did not contain such stimuli for short songs (c, electronic supplementary material, figure S1). (Online version in colour.)

parameters for predicting males' behaviour (figure 2c). Adding even more complexity to the model with the addition of leaky integration or an urgency parameter did not improve performance (electronic supplementary material, table S1). Model parameters were similar for all of the fitted variants of the DDM, indicating that our results are robust to changes in model complexity.

## (b) Males integrate directional cues with long memory, cue-specific weights, low thresholds and high noise

The best fit model indicated that males can integrate directional cues over the whole song, and even in models with a leaky integration, the time constant was estimated at 24 syllables (electronic supplementary material, table S1), which is

twice as long as both the longest song in our dataset (12 syllables; 1.19 s) and a typical female song (12–15 syllables, mean ± s.d. = 1.18 ± 0.23 s; [45]). Thus, sensory information from the whole song has the potential to influence the localization response. The decision threshold $\theta$ of the best fit model had a value of 7.14. Level and timing cues were weighted differently, with the level cue outweighing the timing cue by a factor of 1.65. The minimum number of syllables required to cross the threshold ($\theta/w$) was therefore 8 for timing and 5 for level cues, meaning that the stronger level cues drove faster decisions (figure 2d, electronic supplementary material, figure S4). The low threshold resulted in decisions usually being fixed before the end of long 12-syllable songs (figure 2e), consistent with our finding that the beginning of the song is more and the end of the

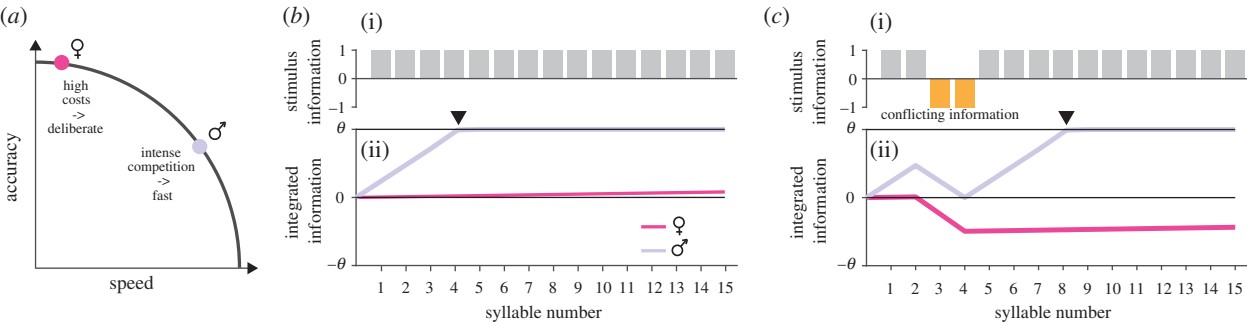

**Figure 3.** Sex-specific speed–accuracy trade-offs arise from differential integration dynamics. (*a*) Males and females differentially resolve speed–accuracy trade-offs when responding to acoustic communication signals. Females (magenta) pay high costs from errors and therefore maximize their accuracy by deliberation at the cost of speed. Males (grey) face intense competition with other males and trade accuracy in favour of speed. (*b, c*) Integration dynamics tune decision making to the sex-specific speed–accuracy trade-offs. Shown is the integrated information (ii) for females (magenta) and males (grey) for stimuli (i) with unequivocal (*b*) and conflicting (*c*) information. Integrated information is scaled relative to the decision thresholds $\theta$ to facilitate the comparison between sexes. For females, individual stimulus elements correspond to species-typical (grey) and untypical (orange) patterns. Species-typical cues have low weight and are not sufficient to fix decisions before song end (*b*). Conflicting (negative) cues have a strong weight and practically veto positive responses (*c*). For males, the stimuli correspond to directional cues. Individual cues have high weight, which accelerates decisions (black arrowhead) for unequivocal information (*b*). Long integration times improve accuracy when cues are conflicting (*c*). See also electronic supplementary material, table S2. Female data from [29]. (Online version in colour.)

song is less influential than expected for observed male turning responses (figure 1*c*, electronic supplementary material, figure S2). This means that males trade accuracy in favour of speed, since responding before the end of song can result in localization errors if sensory information early in the song is unreliable. For the short songs, which largely consisted of the weaker timing cues, sensory information was insufficient to drive decisions by crossing the threshold in our model, and the turning direction was determined by the value of the integrated information at the song end (figure 2*e*). This is consistent with the observation that the average directional cue over the full song is more predictive of behaviour for the short songs, than for the long songs (figure 2*f*). The noise level $\sigma$ of the best fit model was 2.25; thus, the signal-to-noise ratios ($w/\sigma$) were 0.44 for timing cues and 0.73 for level cues. Localization cues provided by single syllables are therefore relatively noisy, and integration is indeed necessary to infer sound direction reliably.

## 4. Discussion

Our DDM of temporal integration applied to male behaviour demonstrates that sexual selection has shaped the neural processing of acoustic stimuli to favour speedy decisions in males, in contrast with the slower, but more accurate decisions in females (figure 3*a*). The model accurately reproduced the males' localization behaviour (figure 2*b,c*) and the model parameters describe an integration process that is consistent with the pressures facing males to rapidly localize a stationary, singing female in a noisy environment (figure 2*d–f*, electronic supplementary material, table S1). The same modelling technique was previously used on females of the same species evaluating songs of potential mates [29], and the differences in model parameters correspond with expectations of sexual selection theory that females should have a higher threshold for response and strongly avoid unattractive signal characteristics (electronic supplementary material, table S2). This is a rare demonstration of variation in temporal integration strategies associated with ecologically relevant and natural behaviours.

### (a) Decisions in males are fast for strong cues and accurate for weak cues

We found that males had a low threshold for response: the average time to decision inferred from the model was much less than the duration of the standard female song stimulus used in this study (figure 2*d,e*). Thus, when the evidence is strong, males can decide quickly. This corresponds with the behaviour of males in localization experiments, in which they frequently turn towards a song before it ends [40]. The best fit model had higher weights for level cues than for timing cues (electronic supplementary material, table S2 and figure S3B), and decisions were therefore faster with level cues than with timing cues (figure 2*d*). This does not imply that level cues always predominate over timing cues; instead the difference likely arose because of the specific values chosen for each cue: the unilateral level cues (resulting in an approximately 8 dB ILD) were expected to provide stronger directional information than the bilateral timing cues (4 ms ITD) [40]. Importantly, this finding implies that cues are weighted by their strength, such that decision making is accelerated when evidence is strong.

Although we expected males' decisions to be biased towards speed, accuracy is also important because mistakes in localization could cause males to move out of hearing range of the female and prevent them from finding one another [35,46]. The signal-to-noise ratio for a single syllable was low (0.44 and 0.73, respectively), and males therefore did integrate over multiple syllables, although they usually reached threshold and made a decision before they heard all of the syllables in the full song (figure 2*d,e*). This also means that males would be less accurate in case directional cues early in the song indicate the wrong direction. Nevertheless, the model indicated that males had the capacity to integrate over a much longer time period if directional cues were weak and the threshold was not reached (electronic supplementary material, table S1). Thus, when directional cues were too weak for a speedy decision, males could integrate additional sensory information, which should improve signal-to-noise ratios and ultimately lateralization accuracy [32]. This explains the high accuracy of male directional responses in the presence of noise [35]. Long integration

times are maladaptive when the information being integrated changes more rapidly than the integration time constant, leading to erroneous decisions [10,47]. However, in this system long integration may have few costs because the information evaluated by males in the female song, her position, is constant prior to the turning decision because females remain stationary while singing. The integration dynamics in males therefore resolve the speed–accuracy trade-off by allowing for flexibility in decision making: sensory information is able to drive fast responses when it is strong, but long integration times allow accurate localization of the female in case of weak cues.

While our experiments were designed to assess the decision-making strategies of males on a population level, some variation in decision-making strategies could depend on male state or consistent differences between individuals. In other species, competitively inferior males use so-called satellite or sneaker strategies to avoid direct competition with dominant males [48]. However, in *C. biguttulus*, direct agonistic interactions between males are rare and the primary means of competition is the ability to rapidly localize females. Slow decision making, similar to females', is therefore unlikely to be an advantageous alternative strategy for males in this species. Given that our model explains the behaviour measured from different sets of males so well ($r^2 = 0.86$, figure 2*b,c*), variation among individuals is likely low, and our main conclusion—that males trade accuracy in favour of speed—is likely to be robust to these factors.

## (b) Integration of courtship signals is tuned to sex-specific costs

Our finding that male *C. biguttulus* have a low threshold for response contrasts with the results from previous studies using a similar behavioural and modelling paradigm to characterize temporal integration in females of the same species [29,30]. Females were tested with songs consisting of a mixture of attractive and unattractive syllables. There was a large difference between males and females in how they weighted sensory information (figure 3). In females, positive cues had a weak influence and on their own could not reach the threshold by the end of the song; in other words, females rarely commit to a positive decision before the end of the song. However, negative cues (i.e. unattractive song syllables) had a much stronger weight and even a few unattractive syllables could reach the threshold for not responding. By contrast, in males, we found that clear directional information had a strong weight and was capable of driving responses before the end of a typical female song. Both sexes had integration times that were longer than the duration of typical songs, but in females, this likely serves less to enhance the signal-to-noise ratio (as we argue is the case for males), but rather to ensure the detection of unattractive elements at any point in the song, preventing them from initiating courtship with a low-quality or heterospecific male.

There are some differences in the behavioural paradigms because females were tested for a response to songs with both positive and negative information on male attractiveness, while males were tested using only attractive syllables but with varied directional cues. Pattern and directional information are extracted from the song in parallel pathways

and the pattern decision then gates turning [36]. Thus, turning in males in this study reflects both the attractiveness of the song syllable and the quality of directional cues. Despite these differences, both the female decision to respond and the male decision to turn signal readiness of each sex to further escalate the courtship interaction. Therefore, the integration differences between males and females reflect differences in the costs and benefits of decision-making strategies affecting each sex. Future studies examining the integration of stimuli with unattractive pattern information in males would further elucidate sex differences in temporal processing in this species.

The neural circuits that integrate directional cues over time to control male turning behaviour are unknown. Peripheral circuits extract directional cues from afferent inputs but do not integrate this information across multiple syllables [49–51]. The evaluation of the song pattern and integration of directional cues is likely to happen in the brain and its results are relayed to the motor centres via descending interneurons [52], but this has not been assessed systematically. In the female brain, auditory activity has been recorded in the lateral protocerebrum, the superior medial protocerebrum and the central complex (CX) [53,54] and electrical stimulation of the CX can elicit the behavioural responses to song in females [55]. In the insect brain, the CX is a central circuit for orientation behaviour with integrator properties [56,57]. It may therefore drive responses also in males and CX neurons themselves or their presynaptic partners may have sex-specific properties that reflect the sex-specific speed–accuracy trade-offs evident from behaviour.

Although the specific neural circuits have not been identified, our DDM is realistic because it replicates identified neural processes. All model parameters map to biophysical properties of decision-making neurons and circuits [58,59]: Sensory weights could correspond to the number and strength of synapses to an integrating neuron. The integration time constant could correspond for instance to the kinetics of intracellular calcium, or to factors that determine the dynamics of a recurrent network [60,61]. The decision threshold could correspond to a spiking threshold determined by the density of sodium channels at the spike initiation zone or controlled by neuromodulators [60,62,63]. Sexual selection could act on these parameters to produce the sex-specific integration of sensory information seen in grasshoppers. Our results therefore point the way towards a study of the evolution of sensory processing mechanisms in realistic ecological contexts and natural behaviours.

**Data accessibility.** The source code for generating the models is available at https://doi.org/10.5281/zenodo.4410311. The behavioural data available from the Dryad Digital Repository: https://doi.org/10.5061/dryad.05qfttf28 [64].

**Authors' contributions.** Experimental design and data collection by B.R. and M.R. Model fitting and analysis by J.C. Manuscript written by J.C., B.R., and M.R.

**Competing interests.** We declare we have no competing interests.

**Funding.** J.C. was funded by DFG CL 596/1-1 (Emmy Noether grant, 32951824), DFG CL 596/2-1 (SPP2205, 430158535); B.R. by DFG Ro 547/12-1; M.R. by US National Science Foundation International Research Fellowship Program (IRFP 1158968).

**Acknowledgements.** Michael Rumpold assisted with the experiments, and members of the Behavioral Physiology Group helped with field collection and husbandry.

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
