## [Peer Review File · Proceedings of the Royal Society B: Biological Sciences]

Review History

RSPB-2020-1971.R0 (Original submission)

Review form: Reviewer 1

Recommendation

Reject – article is scientifically unsound

Scientific importance: Is the manuscript an original and important contribution to its field?

Acceptable

General interest: Is the paper of sufficient general interest?

Acceptable

Quality of the paper: Is the overall quality of the paper suitable?

Marginal

Is the length of the paper justified?

Yes

Should the paper be seen by a specialist statistical reviewer?

No

Do you have any concerns about statistical analyses in this paper? If so, please specify them explicitly in your report.

No

It is a condition of publication that authors make their supporting data, code and materials available - either as supplementary material or hosted in an external repository. Please rate, if applicable, the supporting data on the following criteria.

Is it accessible?

Yes

Is it clear?

Yes

Is it adequate?

Yes

Do you have any ethical concerns with this paper?

No

Comments to the Author

Comments:

1. The authors state that for behavioral decision making, that there is a trade-off between being fast at the risk of being wrong. If the cost of mating attempts for males is low in this species, even if mating with unsuitable females, then is there really a cost and a trade-off, especially in nature? It would be beneficial if the authors were to quantify or describe in a bit more detail the potential costs faced by males for making a wrong decision, to demonstrate an actual trade-off under natural conditions. For example, even if a male were to respond to suboptimal female song in nature, depending on the density of conspecific females in the area, a male can run into suitable females by chance while searching or use other cues for locating correct mates (e.g., visual or olfactory cues). Conversely, it might also be costly for males when they make errors, in a similar way that it is costly for females. Males expose themselves to predators, parasitoids, and rivals when searching for females, with interactions with these groups potentially leading to death, illness, or injury, respectively.

2. More information on the male grasshoppers used in this study is needed to better evaluate the validity and generalizability of the results. For example, the authors only state that mature sexually receptive adult males (line 114 and lines 317-318) were used in trials. The authors do mention that the behavioral data were from a previous study, but the information on these males needs to be included in this current paper. For example, were these males wild-caught or lab-reared? If lab-reared, how many generations? What were the mating histories of these males? Were they unmated or experienced? Did they have previous acoustic experience with female song, e.g., during rearing, that could bias the results? Were males isolated during rearing or reared communally? The type of housing and developmental conditions experienced during pre-trial rearing can strongly affect and bias the behavioral phenotypes of adult individuals assayed in experiments.

3. It is not the case that all males should have the same underlying physiology, especially with respect to female song. For example, given the potential for different male life history and mating strategies in insects, not all males will engage in competition in the same way and equally. There could be a subset of males in the population that are subordinates, sneakers, satellites, or have submissive personalities that alter their behavior and their physiology. For instance, a submissive male can have similar physiological responses as females in the sense that they do not need to be fast, they might be highly selective in what female song that they respond to, and they might be more deliberate in their responses, in order to avoid intense male competition. Here, less competitive males might be more selective in their responses and have higher thresholds for

responses, with the accompanying underlying physiology to facilitate this behavioral phenotype.

Was the potential for variation in male behavioral phenotypes considered? Although the authors varied the signals presented to males, it is also important to examine how variation in the males themselves affects the behavior and physiology of males in response to song.

As in Comment 2, more information on the males needs to be presented. How were males chosen for experiments? What was the age range of males tested? Sexually mature males can vary in their responses whether they are young or old, e.g., younger and less experienced males might be slower to respond to female songs and have higher response thresholds than older and more experienced males. In addition, if male competition is centered on rapidly localizing responsive females, then differences in male resource holding potential (e.g., body size), condition (e.g., health and parasite load), and previous competitive experiences (e.g., winning and loser effects), can affect the behavior and physiology of males, even under laboratory conditions.

4. What were the control stimuli for the playback trials? Was there a negative control stimulus, e.g., to show grasshoppers turning away from the speaker? Was a no-response stimulus also presented to grasshoppers, e.g., a stimulus that grasshoppers would ignore or turn at random to? Could males respond in the same way to closely related heterospecific song presented with the same patterning? Although the grasshoppers do show different responses to the different stimulus patterns, control stimuli need to also be presented to show that the responses of males are aimed towards conspecific female songs. It is possible that the same pattern of results observed in this study can be produced with the songs of heterospecific females or non-specific acoustic pulses with the same level and timing.

5. The authors use “most” in many cases to refer to their data, e.g., the number of males tested in trials and how many stimuli were presented to them (line 321), how many stimuli contained syllables (line 322), and the type of responses produced (line 332). It would be better to provide the actual numbers in the text.

Review form: Reviewer 2

Recommendation

Accept with minor revision (please list in comments)

Scientific importance: Is the manuscript an original and important contribution to its field?

Excellent

General interest: Is the paper of sufficient general interest?

Excellent

Quality of the paper: Is the overall quality of the paper suitable?

Excellent

Is the length of the paper justified?

Yes

Should the paper be seen by a specialist statistical reviewer?

No

Do you have any concerns about statistical analyses in this paper? If so, please specify them explicitly in your report.

No

It is a condition of publication that authors make their supporting data, code and materials available - either as supplementary material or hosted in an external repository. Please rate, if applicable, the supporting data on the following criteria.

Is it accessible?

Yes

Is it clear?

N/A

Is it adequate?

N/A

Do you have any ethical concerns with this paper?

No

Comments to the Author

This is a very interesting and well-written manuscript. Uncharacteristically, I have only a few relatively minor comments, but by and large I strongly recommend this for publication, after minor revisions. The results in favour of the drift diffusion model for explaining the SAT are clear cut (though perhaps mention/briefly explain this model in the abstract?)

Title – would be good to have a specification of taxon.

Abstract: One comment is that the authors could perhaps be a little more upfront about the fact that this is a modelling study, based on previously published behavioural data. There is nothing wrong with that, but one wouldn't easily glean that from looking at the title and abstract.

Methods and results – what's a syllable, and specifically, what defines an attractive or unattractive syllable? This will be obvious to the authors and experts in the field, but for a multidisciplinary journal it would be good to explain these terms and concepts, even if briefly.

Discussion – I would like to see a bit more about where in the brain or peripheral nervous system the neural processing underpinning SATS takes place – this is currently a bit vague, and some ideas, even if speculative, might help here.

Review form: Reviewer 3 (Heiner Römer)

Recommendation

Major revision is needed (please make suggestions in comments)

Scientific importance: Is the manuscript an original and important contribution to its field?

Good

General interest: Is the paper of sufficient general interest?

Excellent

Quality of the paper: Is the overall quality of the paper suitable?

Excellent

Is the length of the paper justified?

Yes

Should the paper be seen by a specialist statistical reviewer?

No

Do you have any concerns about statistical analyses in this paper? If so, please specify them explicitly in your report.

No

It is a condition of publication that authors make their supporting data, code and materials available - either as supplementary material or hosted in an external repository. Please rate, if applicable, the supporting data on the following criteria.

Is it accessible?

Yes

Is it clear?

Yes

Is it adequate?

Yes

Do you have any ethical concerns with this paper?

No

Comments to the Author

MS by Clemens et al: "Sex-specific speed-accuracy tradeoffs shape neural processing of acoustic signals" Submitted to Proceedings of the Royal Society B

The authors used the bidirectional communication of the grasshopper *Chorthippus biguttulus*, where males produce calling songs to attract females, and females respond with their own song, as an ideal model to test predictions about sex-specific differences in temporal integration of sensory cues. In the behavioral experiments, interaural time and level differences were elegantly manipulated in the 12-syllable female song, so that the revealed temporal integration strategies are certainly associated with ecologically relevant conditions. The authors determined a drift-diffusion model with the best fit to the behavioral data for temporal integration in males. The model corresponded very well with decisions in males.

Major point:

Starting in the Abstract, and throughout the text, the authors give the impression that they have studied male behavior, in addition to the modelling approach. For example, line 92: "To test these predictions, we used a two-speaker playback design to measure male localization of artificial female songs with conflicting directional cues." This is not correct, since all behavioral data are taken from Reichert and Ronacher, (2019). Therefore, the distinction between results obtained from the model in the present manuscript and those from behavior published before should be made very clear.

Line 62:enable more accurate decisions about high quality males.

Line 80: ...Clemens et al. 2014; 2017

Line 83: ...with unsuitable, heterospecific females

Line 90: ...from female signals with low signal-to-noise-ratio...

Line 101ff: this whole paragraph deals with M&M, and should be shifted to page 12.

Line 168 – 179: I suggest to shift Table 1 with the results of the model comparisons to Supplementary Material.

Figure 2: to guide the reader's attention specifically to behavioral data, figure 2B and F could be marked with "behavior".

Also, add (+) and (-) to the decision threshold θ in the y-axis of fig. 2A.

Line 226: Does it also conform with the findings of Reichert (2015) on the effect of masking noise on sound localization abilities? Although noise sharply reduced the responsiveness of males to female songs, once males had detected the signal, they responded highly accurately, even at the highest noise levels.

Line 229: ...of temporal integration, and the comparison with male behavior, ...

Line 231: The central message of the MS is the sex-specific difference in the temporal integration of sensory cues, so at this point it may be valuable to present the reader in a new figure this difference in speedy decisions in males, compared to the slower, but more accurate decisions in females. Alternatively, shift Table S1 into the main text.

Line 246: Throughout the text I expected to find values for the latency of the male turning response, which could be directly compared with the data in the DDM, but apparently these data don't exist, except for the more qualitative hint in Helversen and Rheinlaender (1988)?

Line 295: very long sentence.

Line 321: ...with 10 repetitions at a rate of?

Decision letter (RSPB-2020-1971.R0)

02-Nov-2020

Dear Dr Clemens:

I am writing to inform you that your manuscript RSPB-2020-1971 entitled "Sex-specific speed-accuracy tradeoffs shape neural processing of acoustic signals" has, in its current form, been rejected for publication in Proceedings B.

This action has been taken on the advice of referees, who have recommended that substantial revisions are necessary. With this in mind we would be happy to consider a resubmission, provided the comments of the referees are fully addressed. However please note that this is not a provisional acceptance.

Sincerely,

Professor Gary Carvalho
 mailto: proceedingsb@royalsociety.org

Associate Editor

Board Member: 1

Comments to Author:

Thank you for submitting your manuscript to Proceedings B. We have now received three reviews of your manuscript. The reviewers all agree that the study is interesting, but they have a number of concerns that would need to be addressed before the manuscript could be considered for publication in Proceeding B. Two of the reviewers point out that it should be made clear from the start that this is a modelling study with no original behavioral data. The wording in the abstract and throughout the manuscript does not always make this clear. The reviewers also recommend that more information about the study system and the bases for assumptions be included in the manuscript to make the information accessible to readers unfamiliar with the study system. One reviewer was particularly concerned about the experimental design, raising questions about whether individual variation across males was considered and the lack of controls for sound stimuli in the behavioral experiments. This reviewer also questions the assumption that responding to mate signals is necessarily less costly for males than females. The reviewers provide valuable feedback that will improve a future version of the manuscript.

Reviewer(s)' Comments to Author:

Referee: 1

Comments to the Author(s)

Comments:

1. The authors state that for behavioral decision making, that there is a trade-off between being fast at the risk of being wrong. If the cost of mating attempts for males is low in this species, even if mating with unsuitable females, then is there really a cost and a trade-off, especially in nature? It would be beneficial if the authors were to quantify or describe in a bit more detail the potential costs faced by males for making a wrong decision, to demonstrate an actual trade-off under natural conditions. For example, even if a male were to respond to suboptimal female song in nature, depending on the density of conspecific females in the area, a male can run into suitable females by chance while searching or use other cues for locating correct mates (e.g., visual or olfactory cues). Conversely, it might also be costly for males when they make errors, in a similar way that it is costly for females. Males expose themselves to predators, parasitoids, and rivals when searching for females, with interactions with these groups potentially leading to death, illness, or injury, respectively.

2. More information on the male grasshoppers used in this study is needed to better evaluate the validity and generalizability of the results. For example, the authors only state that mature sexually receptive adult males (line 114 and lines 317-318) were used in trials. The authors do mention that the behavioral data were from a previous study, but the information on these males needs to be included in this current paper. For example, were these males wild-caught or lab-reared? If lab-reared, how many generations? What were the mating histories of these males? Were they unmated or experienced? Did they have previous acoustic experience with female song, e.g., during rearing, that could bias the results? Were males isolated during rearing or reared communally? The type of housing and developmental conditions experienced during pre-trial rearing can strongly affect and bias the behavioral phenotypes of adult individuals assayed in experiments.

3. It is not the case that all males should have the same underlying physiology, especially with respect to female song. For example, given the potential for different male life history and mating strategies in insects, not all males will engage in competition in the same way and equally. There could be a subset of males in the population that are subordinates, sneakers, satellites, or have submissive personalities that alter their behavior and their physiology. For instance, a submissive male can have similar physiological responses as females in the sense that they do not need to be fast, they might be highly selective in what female song that they respond to, and they might be more deliberate in their responses, in order to avoid intense male competition. Here, less competitive males might be more selective in their responses and have higher thresholds for responses, with the accompanying underlying physiology to facilitate this behavioral phenotype.

Was the potential for variation in male behavioral phenotypes considered? Although the authors varied the signals presented to males, it is also important to examine how variation in the males themselves affects the behavior and physiology of males in response to song.

As in Comment 2, more information on the males needs to be presented. How were males chosen for experiments? What was the age range of males tested? Sexually mature males can vary in their responses whether they are young or old, e.g., younger and less experienced males might be slower to respond to female songs and have higher response thresholds than older and more experienced males. In addition, if male competition is centered on rapidly localizing responsive females, then differences in male resource holding potential (e.g., body size), condition (e.g., health and parasite load), and previous competitive experiences (e.g., winning and loser effects), can affect the behavior and physiology of males, even under laboratory conditions.

4. What were the control stimuli for the playback trials? Was there a negative control stimulus, e.g., to show grasshoppers turning away from the speaker? Was a no-response stimulus also presented to grasshoppers, e.g., a stimulus that grasshoppers would ignore or turn at random to? Could males respond in the same way to closely related heterospecific song presented with the same patterning? Although the grasshoppers do show different responses to the different stimulus patterns, control stimuli need to also be presented to show that the responses of males are aimed towards conspecific female songs. It is possible that the same pattern of results observed in this study can be produced with the songs of heterospecific females or non-specific acoustic pulses with the same level and timing.

5. The authors use “most” in many cases to refer to their data, e.g., the number of males tested in trials and how many stimuli were presented to them (line 321), how many stimuli contained syllables (line 322), and the type of responses produced (line 332). It would be better to provide the actual numbers in the text.

Referee: 2

Comments to the Author(s)

This is a very interesting and well-written manuscript. Uncharacteristically, I have only a few relatively minor comments, but by and large I strongly recommend this for publication, after

minor revisions. The results in favour of the drift diffusion model for explaining the SAT are clear cut (though perhaps mention/briefly explain this model in the abstract?)

Title – would be good to have a specification of taxon.

Abstract: One comment is that the authors could perhaps be a little more upfront about the fact that this is a modelling study, based on previously published behavioural data. There is nothing wrong with that, but one wouldn't easily glean that from looking at the title and abstract.

Methods and results – what's a syllable, and specifically, what defines an attractive or unattractive syllable? This will be obvious to the authors and experts in the field, but for a multidisciplinary journal it would be good to explain these terms and concepts, even if briefly.

Discussion – I would like to see a bit more about where in the brain or peripheral nervous system the neural processing underpinning SATS takes place – this is currently a bit vague, and some ideas, even if speculative, might help here.

Referee: 3

Comments to the Author(s)

MS by Clemens et al: "Sex-specific speed-accuracy tradeoffs shape neural processing of acoustic signals" Submitted to Proceedings of the Royal Society B

The authors used the bidirectional communication of the grasshopper *Chorthippus biguttulus*, where males produce calling songs to attract females, and females respond with their own song, as an ideal model to test predictions about sex-specific differences in temporal integration of sensory cues. In the behavioral experiments, interaural time and level differences were elegantly manipulated in the 12-syllable female song, so that the revealed temporal integration strategies are certainly associated with ecologically relevant conditions. The authors determined a drift-diffusion model with the best fit to the behavioral data for temporal integration in males. The model corresponded very well with decisions in males.

Major point:

Starting in the Abstract, and throughout the text, the authors give the impression that they have studied male behavior, in addition to the modelling approach. For example, line 92: "To test these predictions, we used a two-speaker playback design to measure male localization of artificial female songs with conflicting directional cues." This is not correct, since all behavioral data are taken from Reichert and Ronacher, (2019). Therefore, the distinction between results obtained from the model in the present manuscript and those from behavior published before should be made very clear.

Line 62: ...enable more accurate decisions about high quality males.

Line 80: ...Clemens et al. 2014; 2017

Line 83: ...with unsuitable, heterospecific females

Line 90: ...from female signals with low signal-to-noise-ratio...

Line 101ff: this whole paragraph deals with M&M, and should be shifted to page 12.

Line 168 – 179: I suggest to shift Table 1 with the results of the model comparisons to Supplementary Material.

Figure 2: to guide the reader's attention specifically to behavioral data, figure 2B and F could be marked with "behavior".

Also, add (+) and (-) to the decision threshold θ in the y-axis of fig. 2A.

Line 226: Does it also conform with the findings of Reichert (2015) on the effect of masking noise on sound localization abilities? Although noise sharply reduced the responsiveness of males to female songs, once males had detected the signal, they responded highly accurately, even at the highest noise levels.

Line 229: ...of temporal integration, and the comparison with male behavior, ...

Line 231: The central message of the MS is the sex-specific difference in the temporal integration of sensory cues, so at this point it may be valuable to present the reader in a new figure this difference in speedy decisions in males, compared to the slower, but more accurate decisions in females. Alternatively, shift Table S1 into the main text.

Line 246: Throughout the text I expected to find values for the latency of the male turning response, which could be directly compared with the data in the DDM, but apparently these data don't exist, except for the more qualitative hint in Helversen and Rheinlaender (1988)?

Line 295: very long sentence.

Line 321: ...with 10 repetitions at a rate of?

Author's Response to Decision Letter for (RSPB-2020-1971.R0)

See Appendix A.

RSPB-2021-0005.R0

Review form: Reviewer 1

Recommendation

Accept as is

Scientific importance: Is the manuscript an original and important contribution to its field?

Good

General interest: Is the paper of sufficient general interest?

Good

Quality of the paper: Is the overall quality of the paper suitable?

Excellent

Is the length of the paper justified?

Yes

Should the paper be seen by a specialist statistical reviewer?

No

Do you have any concerns about statistical analyses in this paper? If so, please specify them explicitly in your report.

No

It is a condition of publication that authors make their supporting data, code and materials available - either as supplementary material or hosted in an external repository. Please rate, if applicable, the supporting data on the following criteria.

Is it accessible?

Yes

Is it clear?

Yes

Is it adequate?

Yes

Do you have any ethical concerns with this paper?

No

Comments to the Author

The manuscript is much improved and the authors have adequately addressed all of the comments of the reviewers.

Review form: Reviewer 3

Recommendation

Accept as is

Scientific importance: Is the manuscript an original and important contribution to its field?

Excellent

General interest: Is the paper of sufficient general interest?

Excellent

Quality of the paper: Is the overall quality of the paper suitable?

Excellent

Is the length of the paper justified?

Yes

Should the paper be seen by a specialist statistical reviewer?

No

Do you have any concerns about statistical analyses in this paper? If so, please specify them explicitly in your report.

No

It is a condition of publication that authors make their supporting data, code and materials available - either as supplementary material or hosted in an external repository. Please rate, if applicable, the supporting data on the following criteria.

Is it accessible?

Yes

Is it clear?

Yes

Is it adequate?

Yes

Do you have any ethical concerns with this paper?

No

Comments to the Author

The authors present a revised manuscript in which all my comments and those of the other two reviewers are followed. They presented more information about the study system, experimental procedures and the bases for their assumptions.

The details on the mating system in *Chorthippus biguttulus*, and how sex-specific costs and errors shape the decision making are particularly useful for readers unfamiliar with the grasshopper communication system.

I like the new Fig. 3 very much as an illustration summary of the sex-specific decision dynamics in females and males. Well done!

Decision letter (RSPB-2021-0005.R0)

19-Jan-2021

Dear Dr Clemens

I am pleased to inform you that your Review manuscript RSPB-2021-0005 entitled "Sex-specific speed-accuracy tradeoffs shape neural processing of acoustic signals in a grasshopper" has been accepted for publication in Proceedings B.

The referee(s) do not recommend any further changes. Therefore, please proof-read your manuscript carefully and upload your final files for publication. Because the schedule for publication is very tight, it is a condition of publication that you submit the revised version of your manuscript within 7 days. If you do not think you will be able to meet this date please let me know immediately.

To upload your manuscript, log into <http://mc.manuscriptcentral.com/prsb> and enter your Author Centre, where you will find your manuscript title listed under "Manuscripts with Decisions." Under "Actions," click on "Create a Revision." Your manuscript number has been appended to denote a revision.

You will be unable to make your revisions on the originally submitted version of the manuscript. Instead, upload a new version through your Author Centre.

1) A text file of the manuscript (doc, txt, rtf or tex), including the references, tables (including captions) and figure captions. Please remove any tracked changes from the text before submission. PDF files are not an accepted format for the "Main Document".

2) A separate electronic file of each figure (tiff, EPS or print-quality PDF preferred). The format should be produced directly from original creation package, or original software format. Please note that PowerPoint files are not accepted.

3) Electronic supplementary material: this should be contained in a separate file from the main text and the file name should contain the author's name and journal name, e.g. `authorname_procb_ESM_figures.pdf`

All supplementary materials accompanying an accepted article will be treated as in their final form. They will be published alongside the paper on the journal website and posted on the online figshare repository. Files on figshare will be made available approximately one week before the accompanying article so that the supplementary material can be attributed a unique DOI. Please see: <https://royalsociety.org/journals/authors/author-guidelines/>

4) Data-Sharing and data citation

It is a condition of publication that data supporting your paper are made available. Data should be made available either in the electronic supplementary material or through an appropriate repository. Details of how to access data should be included in your paper. Please see <https://royalsociety.org/journals/ethics-policies/data-sharing-mining/> for more details.

<http://datadryad.org/submit?journalID=RSPB&manu=RSPB-2021-0005> which will take you to your unique entry in the Dryad repository.

Once again, thank you for submitting your manuscript to Proceedings B and I look forward to receiving your final version. If you have any questions at all, please do not hesitate to get in touch.

Sincerely,

Professor Gary Carvalho

Associate Editor,

Board Member

Comments to Author:

Thank you for addressing all the reviewer comments and for your thorough revision.

Reviewer(s)' Comments to Author:

Referee: 1

Comments to the Author(s).

The manuscript is much improved and the authors have adequately addressed all of the comments of the reviewers.

Referee: 3

Comments to the Author(s).

The authors present a revised manuscript in which all my comments and those of the other two reviewers are followed. They presented more information about the study system, experimental procedures and the bases for their assumptions.

The details on the mating system in *Chorthippus biguttulus*, and how sex-specific costs and errors shape the decision making are particularly useful for readers unfamiliar with the grasshopper communication system.

I like the new Fig. 3 very much as an illustration summary of the sex-specific decision dynamics in females and males. Well done!

Decision letter (RSPB-2021-0005.R1)

21-Jan-2021

Dear Dr Clemens

I am pleased to inform you that your manuscript entitled "Sex-specific speed-accuracy tradeoffs shape neural processing of acoustic signals in a grasshopper" has been accepted for publication in Proceedings B.

Open Access

Paper charges

Sincerely,
Proceedings B
<mailto:proceedingsb@royalsociety.org>

Appendix A

Reviewer's comments are marked in blue font, our reply in black font.

Text cited from the revised manuscript is italicized.

All changes to the text are marked in red in a doc "main_changes.docx" submitted with the revision. Pages and line numbers refer to that document.

Board Member: 1

Comments to Author:

Thank you for submitting your manuscript to Proceedings B. We have now received three reviews of your manuscript. The reviewers all agree that the study is interesting, but they have a number of concerns that would need to be addressed before the manuscript could be considered for publication in Proceeding B.

We are happy that all referees found the manuscript interesting. Based on the constructive comments of all three referees, we have thoroughly revised the manuscript, with new sections in introduction and methods, edits for clarity throughout, and as well as a new figure that summarizes our main result. We also moved the Methods section to the beginning of the manuscript to comply with the standards of manuscript formatting at Proceedings B.

Two of the reviewers point out that it should be made clear from the start that this is a modelling study with no original behavioral data. The wording in the abstract and throughout the manuscript does not always make this clear.

We apologize for not making clear the source of the data in the original manuscript. Data for more than half of the stimuli (43/81) are new and unpublished, while 38/81 stimuli were previously published in Reichert and Ronacher (2019). This is now explicitly stated in the introduction and the methods and we clarified wording throughout the manuscript. For instance:

Introduction (p3, l67):

Here we combine an existing behavioral data set [23] with new data and fit a drift-diffusion model [5] to characterize how the nervous system accumulates sensory cues and triggers decisions in mate searching.

Introduction (p4, l113):

To test these predictions, we used new and previously published behavioral data from a two-speaker playback design that measured male localization of artificial female songs with conflicting directional cues [23].

Methods, (p5, l160):

Data for 38/81 stimuli were previously published in [23].

The reviewers also recommend that more information about the study system and the bases for assumptions be included in the manuscript to make the information accessible to readers unfamiliar with the study system. One reviewer was particularly concerned about the experimental design, raising questions about whether individual variation across males was considered and the lack of controls for sound stimuli in the behavioral experiments. This reviewer also questions the assumption that responding to mate signals is necessarily less costly for males than females. The reviewers provide valuable feedback that will improve a future version of the manuscript.

Following the referees' suggestions, we added new sections to the introduction with more details on the study system to better explain our assumptions, for instance, regarding relative costs for males and females. In addition, more details on the animals and on the experimental procedures (including the control stimuli) are now given in Methods. Please see below for our detailed responses to these comments.

Reviewer(s)' Comments to Author:

Referee: 1

Comments to the Author(s)

Comments:

1. The authors state that for behavioral decision making, that there is a trade-off between being fast at the risk of being wrong. If the cost of mating attempts for males is low in this species, even if mating with unsuitable females, then is there really a cost and a trade-off, especially in nature? It would be beneficial if the authors were to quantify or describe in a bit more detail the potential costs faced by males for making a wrong decision, to demonstrate an actual trade-off under natural conditions.

This is an excellent point. We now provide more details on the mating system in *Chorthippus biguttulus* that clarify how sex-specific costs and errors shape the decision making. We will first reply to the referee's individual comments and suggestions. These points are summarized in new sections in the introduction (p3, l79), which is provided at the end of our reply to this comment.

Our working hypothesis was that sex-specific costs of being wrong and slow should shape how cues from courtship signals are integrated in both sexes in *C. biguttulus*: that females need to be correct and males need to be fast.

Females pay high potential costs from responding to inappropriate males, since females are egg-limited and expose themselves to predators and parasitoids when singing. Moreover, females will waste time interacting with males that they are unlikely to mate with. Females should therefore avoid responding to and interacting with males of low quality. Females can infer male quality from the song - for instance, males from a different species, males with developmental abnormalities, or with injuries produce songs with "gappy" syllables, which females do not respond to (Kriegbaum 1989, Clemens et al. 2017). This female choosiness for male song pattern is known from multiple studies in the system (e.g. Kriegbaum 1989, Helversen and Helversen 1994). On the other hand, females do not need to be fast when evaluating the pattern of the male song, because there is no strong competition between females to attract "good" males.

We have previously shown that this is reflected in how females integrate pattern information from the song. They trade speed in favor of accuracy: females heavily weigh negative cues - e.g. gappy syllables - found in the song of males from other species, or conspecific males with developmental defects or injuries. And they delay their decision to sing until they have evaluated all syllables of a song - to detect any evidence that the male is of low quality.

In this study, we designed experiments to test how males trade speed and accuracy when integrating *directional* cues from the female song. Being slow and wrong when localizing females is costly in males and they therefore have to resolve a speed and accuracy trade-off differently than females. Localization errors will make males move in the wrong direction. In the worst case, the male will fail to reach the female. In the best case, errors will delay the male's approach, which will in turn make males lose the race to the female and extend their exposure to predators and parasites. Males should therefore not completely sacrifice accuracy. But they are still expected to be faster than females. This is because intense competition with other males to be the first to localize a female pushes males to be fast. Females cease producing response songs once engaged in close-range courtship with another male and therefore only the first male arriving at the female will be able to court and possibly mate.

According to our hypothesis, males should therefore resolve the speed-accuracy trade-off when integrating directional cues by favoring speed over accuracy: Males should respond as soon as they have sufficient directional information. This will maximize speed, though at the cost of reduced localization accuracy.

For example, even if a male were to respond to suboptimal female song in nature, depending on the density of conspecific females in the area, a male can run into suitable females by chance while searching or use other cues for locating correct mates (e.g., visual or olfactory cues).

Our experiments were not designed to test whether or not males should respond to a suboptimal female song, but how they integrate potentially noisy directional cues from an attractive female song - our hypothesis was that males should not deliberate too long before deciding where to turn to because they face competition from other males.

The fast and accurate evaluation of directional cues from female song is crucial, because the probability of encountering a receptive female is low otherwise (Kriegbaum & Helversen 1992): density of receptive females is low, the habitat is visually cluttered and females do not emit long-range chemical cues. In addition, a female encountered by chance will likely be not receptive, while a female that responds certainly will be.

Conversely, it might also be costly for males when they make errors, in a similar way that it is costly for females. Males expose themselves to predators, parasitoids, and rivals when searching for females, with interactions with these groups potentially leading to death, illness, or injury, respectively.

This is correct. Males do expose themselves to predators and parasitoids when singing and jumping. And too many localization errors will make males miss the female. This is now made explicit in the introduction (see below, p4, I102).

On the other hand, exposure to rival males is unlikely to be costly in this system. Mate competition in this species involves rapid localization of receptive females and direct aggressive interactions are not observed.

The above points are now summarized in new sections in the introduction (p3, l79):
In C. biguttulus, females pay high costs from making errors [25,27,28], because they are egg-limited and mating with a male of another species or of low genetic quality produces no or low-quality offspring. In addition, singing exposes females to predators and parasitoids. Females should therefore avoid responding to the song from males of another species or of low genetic quality. By contrast, females do not face competition from other females and therefore are not under pressure to be fast. Females are therefore expected to favor accuracy over speed when evaluating the male song pattern. This was confirmed by a drift-diffusion model for temporal integration based on female response behavior [29,30]. The model parameters indicated that females integrate information across the entire calling song of a male with a high threshold for response and very high negative weighting of unattractive song components (i.e., those of heterospecific or malformed males). These integration parameters ensure the accurate detection of unsuitable males combined with slow behavioral responses to attractive males.

Here, we extend the modelling approach to male mate localization, which is predicted to have very different integration characteristics. Female density is low in the visually cluttered environment and females lack conspicuous visual characteristics or long-range chemical cues that would allow males to find them. Chance encounters are therefore rare and the female response song is often the only possibility for localizing receptive females [28,31]. Furthermore, the speed of approach is critical because females already engaged in close-range courtship with faster arriving males will not continue to advertise their position, preventing slower males from localizing those females. Thus, although males rarely directly interact with one another in physical competitions, they nevertheless face high levels of competition to rapidly localize responsive females in a crowded and noisy environment [32]. Males should therefore favor speed over accuracy to a greater degree than females, although we do not expect males to completely disfavor accuracy when integrating directional cues from the female song, since localization errors increase the time exposed to predators and parasitoids, and will prevent them from finding the female. We predict therefore that males will have higher sensory weightings (or equivalently, a lower response threshold) than females, reflecting their speedier response. This puts them at risk of making errors if early sensory information is wrong. Furthermore, to maintain accuracy when directional cues are equivocal, we predict that integration times will be at least as long as typical female songs so that males can maximize the chances of integrating sufficient directional information from female signals.

2. More information on the male grasshoppers used in this study is needed to better evaluate the validity and generalizability of the results. For example, the authors only state that mature sexually receptive adult males (line 114 and lines 317-318) were used in trials. The authors do mention that the behavioral data were from a previous study, but the information on these males needs to be included in this current paper. For example, were these males wild-caught or lab-reared? If lab-reared, how many generations? What were the mating histories of these males? Were they unmated or experienced? Did they have previous acoustic experience with female song, e.g., during rearing, that could bias the results? Were males isolated during rearing or reared communally? The type of

housing and developmental conditions experienced during pre-trial rearing can strongly affect and bias the behavioral phenotypes of adult individuals assayed in experiments.

We apologize for this omission. More information is now provided in a new section in Methods (p5, l125):

Animals

Behavioral data were collected as described in [23]. We used lab-reared and wild-caught males. Lab-reared males were the F1 offspring from wild-caught individuals and were isolated by sex at the last instar nymph stage and reared in cages separated by sex. Wild-caught males could have mated previously but were kept separate from females for at least three days before the experiments, which is sufficient for them to regain motivation to respond to female signals [33,34]. To further minimize variation in male motivation to respond, we only tested males that responded to a test signal from an attractive female, indicating high motivation to engage in courtship behavior. We did not systematically track male age or exclude males based on their age. Both lab-reared and wild-caught males were group-housed in male-only cages in similar conditions. In group housing, all males would have been exposed to the song of other males, as well as the songs of female playbacks used to assess male motivation and identify test subjects. We, as well as previous studies [35], did not observe any systematic differences in the selectivity of lab-reared or wild-caught males, nor any effect of previous experience on the integration of acoustic cues from females. There is also no evidence for learning in this communication system.

3. It is not the case that all males should have the same underlying physiology, especially with respect to female song. For example, given the potential for different male life history and mating strategies in insects, not all males will engage in competition in the same way and equally. There could be a subset of males in the population that are subordinates, sneakers, satellites, or have submissive personalities that alter their behavior and their physiology. For instance, a submissive male can have similar physiological responses as females in the sense that they do not need to be fast, they might be highly selective in what female song that they respond to, and they might be more deliberate in their responses, in order to avoid intense male competition. Here, less competitive males might be more selective in their responses and have higher thresholds for responses, with the accompanying underlying physiology to facilitate this behavioral phenotype.

We agree that male-male variability is an important factor and there likely exists behavioral variability between males. However, from the many previous studies in this system there is no indication at all of divergent male strategies, like the existence of sneaker or satellite males. Moreover, given the mating system in *C. biguttulus*, which we now describe in more detail in the introduction (p3, l79, see above), such divergent strategies are unlikely. The primary means of competition is the ability to rapidly localize females. Females respond to male calls from a stationary position; males are the ones performing the mate localization. The only way for a male to get a mate is to localize the female. A satellite strategy could therefore not work in this species because the females are stationary. A sneaker male would have to closely track a dominant male but will be unlikely to arrive first at the female.

Lastly, submissive males could avoid potentially high male-male competition by being more deliberate. However, there is no evidence for such behaviour in *C. biguttulus*, since male-male competition largely exists at the level of the race to the female; direct male-male

competition during close-range courtship towards a female is rare. Therefore, more deliberate decision making to avoid direct male-male competition is unlikely to be advantageous, because a more deliberate male will likely lose the race to a receptive female.

In summary, differences between males are highly unlikely to manifest as fundamentally different strategies for resolving the speed-accuracy trade-off when localizing a female, but may play a role at a more subtle level of the fine-tuning of the speed-accuracy tradeoff (to which, see below).

Was the potential for variation in male behavioral phenotypes considered? Although the authors varied the signals presented to males, it is also important to examine how variation in the males themselves affects the behavior and physiology of males in response to song.

Our current study is a population level analysis of sex-specific differences in decision making strategies - our experiments were not designed to assess consistent inter-individual differences in male strategies. Nevertheless, we did not observe consistent differences in male responses if directional cues were unequivocal. For instance, the turns of all males for a stimulus that was broadcast from only one speaker stimulus were directed towards the broadcasting speaker.

Lastly, the existence of consistent individual level differences, would not contradict our conclusion of differences - at the population level, males do indeed favor speed over accuracy when integrating directional cues from females. It would be very interesting to design experiments that analyze male decision-making strategies at the individual level, to assess whether different males idiosyncratically resolve the speed-accuracy trade-off. We think this would be an interesting theme for a future study.

As in Comment 2, more information on the males needs to be presented. How were males chosen for experiments? What was the age range of males tested? Sexually mature males can vary in their responses whether they are young or old, e.g., younger and less experienced males might be slower to respond to female songs and have higher response thresholds than older and more experienced males. In addition, if male competition is centered on rapidly localizing responsive females, then differences in male resource holding potential (e.g., body size), condition (e.g., health and parasite load), and previous competitive experiences (e.g., winning and loser effects), can affect the behavior and physiology of males, even under laboratory conditions.

We now added the requested details to a new section in Methods ("Animals", p5, l125, see reply to comment 2 above). Briefly, males were chosen based on their motivation to court, that is, only if they responded to a test signal from an attractive female. Since *C. biguttulus* males will produce calling songs throughout their lifetime, we did not systematically track male age or exclude males below or above a certain age (Heinrich 2012, Wirmer 2010). We controlled for motivation using two methods: First, by only testing sexually mature males that were motivated to respond to female songs. Second, by isolating animals by sex after the last instar nymph stage (lab reared) or for at least 3 days (wild-caught). The 3-day period is motivated by experimental findings that show that male motivation to court saturates after 3 days of isolation from females (Wirmer et al. 2010).

There is only very rare agonistic competition between males in this species. Resource holding potential and winner/loser effects are not likely to play a strong role. The “goal” of males is to *localize* a receptive female. While there may be some underlying variation in how males do this due to physiology, we do not consider it likely that selection would have favored some males to have a slower response because of their competitive ability.

If these factors played a decisive role, we would expect that our model performance – which assumes a single decision-making strategy across all males in the experimental population – should be much worse. But the performance of our model, based on data from stimuli that were tested with different sets of males, is very good ($r^2=0.86$).

Further dissecting the effects of age, physiological state, and previous experience would be interesting. In the future, it will be interesting to systematically alter male state and for instance compare decision making strategies in receptive males of different ages, or males that have been recently mated or not. However, for our present population-level study our inferences on differences between the sexes remain valid.

The above points are now summarized in a new section in the discussion (p13, l419):
While our experiments were designed to assess the decision-making strategies of males on a population level, some variation in decision-making strategies could depend on male state or consistent differences between individuals. In other species, competitively inferior males use so-called satellite or sneaker strategies to avoid direct competition with dominant males [48]. However, in C. biguttulus direct agonistic interactions between males are rare and the primary means of competition is the ability to rapidly localize females. Slow decision making, similar to females', is therefore unlikely to be an advantageous alternative strategy for males in this species. Given that our model explains the behavior measured from different sets of males so well ($r^2=0.86$, Fig. 2 B, C), variation among individuals is likely low, and our main conclusion – that males trade accuracy in favor of speed – is likely to be robust to these factors.

4. What were the control stimuli for the playback trials? Was there a negative control stimulus, e.g., to show grasshoppers turning away from the speaker? Was a no-response stimulus also presented to grasshoppers, e.g., a stimulus that grasshoppers would ignore or turn at random to? Could males respond in the same way to closely related heterospecific song presented with the same patterning? Although the grasshoppers do show different responses to the different stimulus patterns, control stimuli need to also be presented to show that the responses of males are aimed towards conspecific female songs. It is possible that the same pattern of results observed in this study can be produced with the songs of heterospecific females or non-specific acoustic pulses with the same level and timing.

Males analyze the female song pattern, and only turn if the pattern matches the conspecific song. Since our focus was on how directional cues are integrated to localize females, not on species discrimination, all stimuli have an “attractive” pattern that mimics the female conspecific response song which is known to reliably trigger male responses.

Stimuli differed only in the amount and order of conflicting directional cues, which allowed us to assess how much directional information males require before they approach a female. Our stimulus set contained two control stimuli. First, males reliably turned towards a stimulus with unequivocal directional cues: 90% of trials with such a stimulus elicited turns, 100% of these turns were correct. Second, a stimulus with neutral directional cues that males are expected to turn at random to elicited random turning behavior (0.53, expected value is 0.5 for random directional responses). This is consistent with our model which predicts random decisions if males detect a female song, but get no directional cues from it.

These two control stimuli are now mentioned in Methods (p6, l193):

A control stimulus with neutral directional cues elicited turning responses with random directions (score 0.53, random turning would produce 0.5). Another control stimulus that was broadcast from only one speaker, reliably elicited turns in males (90% of trials) and all of those turns were correctly directed towards the broadcasting speaker.

We did not test different female song patterns (e.g., those of other species) because this was not a study of species discrimination or signal recognition (the stimulus characteristics associated with species discrimination have been the subject of many previous studies in this species). A non-attractive song (or any random sound) will simply fail to trigger a male turning response. For instance, patterns that lack a syllable pause-structure or otherwise deviate too strongly from the female-typical pattern rarely elicit turning responses. However, if they do these turns are correctly directed towards the single speaker (i.e., they are still localizable; for instance, Ronacher & Krahe 1998, Helversen & Helversen 1995, Helversen & Helversen 1997). To our knowledge there is no stimulus that would reliably cause the male to turn away from the speaker.

The pattern of the female song and what constitutes attractive and unattractive stimuli is now explained in Methods (p5, 144):

A female song consists of subunits (“syllables”) that are separated by pauses. The syllables in our female model song were separated by a 17.5 ms pause; each syllable consisted of 6 sound pulses (average pulse duration 10.7 ms). This stimulus pattern was highly attractive and reliably elicited turning responses in males, allowing us to assess how directional cues from the stimulus were integrated by the males. Syllables that lack a pause or do not consist of distinct sound pulses are not attractive to males and fail to elicit male turning responses [36-38].

The fact that males reliably respond to stimuli with unequivocal directional cues reflects the high weight of pattern information and is direct evidence males’ decision-making strategies differ from those of females, which put only small weight on “attractive” male patterns. This difference is now illustrated in a new Fig. 3. In the discussion, we mention that future studies will more precisely assess how pattern cues of different valence are integrated in males (p14, l456).

5. The authors use “most” in many cases to refer to their data, e.g., the number of males tested in trials and how many stimuli were presented to them (line 321), how many stimuli contained syllables (line 322), and the type of responses produced (line 332). It would be better to provide the actual numbers in the text.

We are now more precise in the methods: p5, line 151:

Stimuli were broadcast at 60 dB SPL at the position of the male. Males were presented with ten repetitions of each stimulus. We tested a total of 204 males and most males were tested with more than one stimulus. The median number of different stimuli tested per male (with 10 repetitions per stimulus) was 8 (inter-quartile range 5-10). Stimuli were repeated at a variable rate because each time the male moved, we had to re-position the speakers to center the male once he was again stationary. 44 stimuli contained 12 syllables to mimic a typical female song, but we also tested shortened stimuli with 8 (3 stimuli), 5 (33 stimuli) or 3 (1 stimulus) syllables to better characterize the dynamics of integration. We include data for male responses from a total of 81 stimuli (Fig. S1). Data for 38/81 stimuli were previously published in [23].

Referee: 2

Comments to the Author(s)

This is a very interesting and well-written manuscript. Uncharacteristically, I have only a few relatively minor comments, but by and large I strongly recommend this for publication, after minor revisions. The results in favour of the drift diffusion model for explaining the SAT are clear cut (though perhaps mention/briefly explain this model in the abstract?)

Title – would be good to have a specification of taxon.

We change the title to: “Sex-specific speed-accuracy tradeoffs shape neural processing of acoustic signals in a grasshopper”. The species name is mentioned in the abstract as is the specific type of model.

Abstract: One comment is that the authors could perhaps be a little more upfront about the fact that this is a modelling study, based on previously published behavioural data. There is nothing wrong with that, but one wouldn't easily glean that from looking at the title and abstract.

This was also brought up by board member 1 and referee 3.

The following is copied from our reply to this point above:

We apologize for not making clear the source of the data in the original manuscript. Data for more than half of the stimuli (43/81) are new and unpublished, while 38/81 stimuli were previously published in Reichert and Ronacher (2019). This is now explicitly stated in the introduction and the methods and we clarified wording throughout the manuscript. For instance:

Introduction (p3, l67):

Here we combine an existing behavioral data set [23] with new data and fit a drift-diffusion model [5] to characterize how the nervous system accumulates sensory cues and triggers decisions in mate searching.

Introduction (p4, l113):

To test these predictions, we used new and previously published behavioral data from a two-speaker playback design that measured male localization of artificial female songs with conflicting directional cues [23].

Methods, (p5, l160):

Data for 38/81 stimuli were previously published in [23].

Methods and results – what’s a syllable, and specifically, what defines an attractive or unattractive syllable? This will be obvious to the authors and experts in the field, but for a multidisciplinary journal it would be good to explain these terms and concepts, even if briefly.

The meaning of these terms is now clarified throughout the text:

“Syllable” is now defined at the first mention in the results (p5, l144):

Female song consists of trains of subunits (“syllables”) interleaved by brief pauses.

The meaning of “attractive syllable” is now clarified in the introduction (p4, l89 ...*unattractive song components (i.e., those of heterospecific or malformed males)* and further explained in Methods, p13, l554:

A female song consists of subunits (“syllables”) that are separated by pauses. The syllables in our female model song were separated by a 17.5 ms pause; each syllable consisted of 6 sound pulses (average pulse duration 10.7 ms). This stimulus pattern was highly attractive and reliably elicited turning responses in males, allowing us to assess how directional cues from the stimulus were integrated by the males. Syllables that lack a pause or do not consist of distinct sound pulses are not attractive to males and fail to elicit male turning responses [36-38].

Discussion – I would like to see a bit more about where in the brain or peripheral nervous system the neural processing underpinning SATS takes place – this is currently a bit vague, and some ideas, even if speculative, might help here.

We added the following to the discussion (p14, l459):

The neural circuits that integrate directional cues over time to control male turning behavior are unknown. Peripheral circuits extract directional cues from afferent inputs but do not integrate this information across multiple syllables [49-51]. The evaluation of the song pattern and integration of directional cues is likely to happen in the brain and its results are relayed to the motor centers via descending interneurons [52], but this has not been assessed systematically. In the female brain, auditory activity has been recorded in the lateral protocerebrum, the superior medial protocerebrum and the central complex (CX) [53,54] and electrical stimulation of the CX can elicit the behavioral responses to song in females [55]. In the insect brain, the CX is a central circuit for orientation behavior with integrator properties [56,57]. It may therefore drive responses also in males and CX neurons themselves or their presynaptic partners may have sex-specific properties that reflect the sex-specific speed-accuracy trade-offs evident from behavior.

Referee: 3

Comments to the Author(s)

MS by Clemens et al: “Sex-specific speed-accuracy tradeoffs shape neural processing of acoustic signals” Submitted to Proceedings of the Royal Society B

The authors used the bidirectional communication of the grasshopper *Chorthippus biguttulus*, where males produce calling songs to attract females, and females respond with their own song, as an ideal model to test predictions about sex-specific differences in temporal integration of sensory cues. In the behavioral experiments, interaural time and level differences were elegantly manipulated in the 12-syllable female song, so that the revealed temporal integration strategies are certainly associated with ecologically relevant conditions. The authors determined a drift-diffusion model with the best fit to the behavioral data for temporal integration in males. The model corresponded very well with decisions in males.

Major point:

Starting in the Abstract, and throughout the text, the authors give the impression that they have studied male behavior, in addition to the modelling approach. For example, line 92: “To test these predictions, we used a two-speaker playback design to measure male localization of artificial female songs with conflicting directional cues.” This is not correct, since all behavioral data are taken from Reichert and Ronacher, (2019). Therefore, the distinction between results obtained from the model in the present manuscript and those from behavior published before should be made very clear.

This was also brought up by board member 1 and referee 2.

The following is copied from our reply to this point above:

We apologize for not making clear the source of the data in the original manuscript. Data for more than half of the stimuli (43/81) are new and unpublished, while 38/81 stimuli were previously published in Reichert and Ronacher (2019). This is now explicitly stated in the introduction and the methods and we clarified wording throughout the manuscript. For instance:

Introduction (p3, l67):

Here we combine an existing behavioral data set [23] with new data and fit a drift-diffusion model [5] to characterize how the nervous system accumulates sensory cues and triggers decisions in mate searching.

Introduction (p4, l113):

To test these predictions, we used new and previously published behavioral data from a two-speaker playback design that measured male localization of artificial female songs with conflicting directional cues [23].

Methods, (p5, l160):

Data for 38/81 stimuli were previously published in [23].

Line 62:enable more accurate decisions about high quality males.

Changed to: ...enable more accurate decisions about male quality.

Line 80: ...Clemens et al. 2014; 2017

Done

Line 83: ...with unsuitable, heterospecific females

Done

Line 90: ...from female signals with low signal-to-noise-ratio...

Done

Line 101ff: this whole paragraph deals with M&M, and should be shifted to page 12.

We have now moved the Methods section and removed redundant text from Results.

Line 168 – 179: I suggest to shift Table 1 with the results of the model comparisons to Supplementary Material.

Done

Figure 2: to guide the reader's attention specifically to behavioral data, figure 2B and F could be marked with "behavior".

We marked 2B with "behavior vs. model" and 2F with "r² behavior-stimulus".

Also, add (+) and (-) to the decision threshold θ in the y-axis of fig. 2A.

Done

Line 226: Does it also conform with the findings of Reichert (2015) on the effect of masking noise on sound localization abilities? Although noise sharply reduced the responsiveness of males to female songs, once males had detected the signal, they responded highly accurately, even at the highest noise levels.

We now raise this excellent point in the discussion (p13, l406):

Thus, when directional cues were too weak for a speedy decision, males could integrate additional sensory information, which should improve signal-to-noise ratios and ultimately lateralization accuracy [32]. This explains the high accuracy of male directional responses in the presence of noise [35].

Line 229: ...of temporal integration, and the comparison with male behavior, ...

Done

Line 231: The central message of the MS is the sex-specific difference in the temporal integration of sensory cues, so at this point it may be valuable to present the reader in a new figure this difference in speedy decisions in males, compared to the slower, but more accurate decisions in females. Alternatively, shift Table S1 into the main text.

A new Fig. 3 illustrates how sex-specific decision dynamics tune SATs in females and males.

Line 246: Throughout the text I expected to find values for the latency of the male turning response, which could be directly compared with the data in the DDM, but apparently these data don't exist, except for the more qualitative hint in Helversen and Rheinlaender (1988)? Latency values would be a highly valuable additional data point for assessing decision making strategies. However, these data could not be obtained in the original study. Older studies estimate reaction times to about 500ms in males for stimuli with unequivocal

directional cues and attractive syllables (Ronacher et al. 2000 JCPA, D. von Helversen, 1997 book article).

We now explicitly address the lack of latency measurements in the methods (p6, l169): *The experimental setup did not allow us to score turning latencies and those data were therefore not available for model fitting. However, our stimulus design, with conflicting cues placed in different positions within the song, allows us to reliably infer the dynamics of cue integration from the response scores (see below).*

Line 295: very long sentence.

Thanks – the sentence is now split.

Line 321: ...with 10 repetitions at a rate of?

Repetition rate was under manual control and therefore somewhat variable (we had to re-position the speaker after each male movement). This is now clarified in Methods (p5, l155): *Stimuli were repeated at a variable rate because each time the male moved, we had to re-position the speakers to center the male once he was again stationary.*